Nectar robbing by the invasive bumblebee Bombus terrestris (Apidae) changes the behavior of native flower visitors of Fuchsia magellanica Lam. (Onagraceae) but not seed set

Valdivia Carlos E. carlos.valdivia@ulagos.cl 1
Orellana José I. 2 3
Murúa Maureen 4
1 Laboratorio de Vida Silvestre, Departamento de Ciencias Biológicas y Biodiversidad, Universidad de Los Lagos , Osorno , Región de Los Lagos , Chile
2 Instituto de Biología, Facultad de Ciencias, Pontificia Universidad Católica de Valparaíso , Valparaíso , Región de Valparaíso , Chile
3 Millenium Nucleus of Patagonian Limit of Life (LiLi) , Valdivia , Región de Los Ríos , Chile
4 Centro GEMA: Genómica, Ecología y Medio Ambiente, Universidad Mayor , Santiago , Región Metropolitana , Chile
Manjarrez Javier
Electronic publication date: 2025 Oct 22
Publication date: 2025
Volume: 13
Electronic Location ID: e20253
Received 2025 Jul 24; Accepted 2025 Sep 25
Copyright: ©2025 Valdivia et al.
Copyright year: 2025
Copyright holder: Valdivia et al.
License: This is an open access article distributed under the terms of the Creative Commons Attribution License, which permits unrestricted use, distribution, reproduction and adaptation in any medium and for any purpose provided that it is properly attributed. For attribution, the original author(s), title, publication source (PeerJ) and either DOI or URL of the article must be cited.
License URL: https://creativecommons.org/licenses/by/4.0/

Keywords: Invasive species, Foraging behavior, Hummingbirds, Pollination, Temperate forests, Chile, Sephanoides sephaniodes, Bombus dahlbomii, Apis mellifera

Funding: Fondecyt Initiation into Research 11110230 (CEV) and Regular Internal Research Project R07/18 from the Dirección de Investigación, Universidad de Los Lagos This study was funded by Fondecyt Initiation into Research 11110230 (CEV) and Regular Internal Research Project R07/18 from the Dirección de Investigación, Universidad de Los Lagos. The funders had no role in study design, data collection and analysis, decision to publish, or preparation of the manuscript.

==============================
Mutualisms between plants and pollinators may be threatened by nectar robbers, as these animals typically consume nectar without providing the essential pollination service. In the temperate forests of Chile, the shrub Fuchsia magellanica is primarily pollinated by the hummingbird Sephanoides sephaniodes and the native bumblebee Bombus dahlbomii. However, some populations are also affected by nectar robbing from the invasive bumblebee Bombus terrestris. In a study evaluating 24 populations of F. magellanica, we assessed the effects of nectar robbing on nectar production, the foraging behavior of key pollinators, and the reproductive success of the plant. We measured the nectar production rate and standing crop in flowers that were either protected from visitors or exposed to them. We also observed flower visitation, the foraging behavior of pollinators, and seed sets in experimentally manipulated flowers. Results indicated that nectar robbing reduced nectar production rates by a factor of 4, and the standing crop of nectar by 2.9 times. With an increased percentage of flowers pierced by B. terrestris, the hummingbird S. sephaniodes appeared to reduce its pollinating visits, while B. terrestris increased its robbing visits. Despite these changes in pollinator behavior, there was no significant effect on the seed set. Ultimately, our findings suggest that for F. magellanica, B. terrestris does not pose a significant threat to reproduction, despite its disruptive impact on pollinator behavior.

Introduction

Pollinators frequently visit flowers, contacting the reproductive structures and enabling fertilization. However, not all flower visitors are legitimate pollinators. Some, known as primary nectar robbers, make incisions in the corolla to collect nectar without entering the flower itself (Inouye, 1980). Additionally, these holes can sometimes be exploited by other species, termed secondary nectar robbers (Inouye, 1980). In both cases, nectar robbers typically extract nectar without providing any pollination services, which negatively impacts plant species in at least two ways (Maloof & Inouye, 2000; Irwin et al., 2010). Firstly, nectar robbers can directly damage the fertile parts of flowers, including ovaries and stamens, which can reduce the plant’s maternal function and/or its ability to sire offspring (Maloof & Inouye, 2000; González-Gómez & Valdivia, 2005; Irwin et al., 2010). Secondly, they may indirectly decrease overall seed production by reducing nectar production rates and the standing crop, which makes flowers less attractive to legitimate pollinators (Maloof & Inouye, 2000; González-Gómez & Valdivia, 2005; Irwin et al., 2010).

Indirect effects of nectar robbing can lead to varying outcomes for plants, depending on how legitimate pollinators respond to the robbed flowers (Navarro, 2000). In some cases, a reduction in nectar volume can increase nectar viscosity, which may reduce foraging efficiency and cause pollinators to avoid those flowers (Zhang, Zhao & Inouye, 2014). Alternatively, pollinators may expand their foraging area, increasing their flight distances between plants. This can reduce geitonogamy, i.e., pollination between flowers of the same plant, and promote outcrossing, i.e., pollination between different plants (Irwin et al., 2010). However, changes in visitation patterns alone are not always reliable indicators of plant reproductive success, because pollinator visits do not necessarily result in successful pollen transfer or fertilization (Maloof & Inouye, 2000; Irwin et al., 2010). For this reason, evaluating reproductive outcomes, such as seed set, is essential to fully understand the net effects of nectar robbing (Maloof & Inouye, 2000; Irwin et al., 2010). The response of pollinators to nectar robbing often depends on the species involved, as some taxonomic groups are more susceptible than others, leading to different reproductive impacts. For example, Hazlehurst & Karubian (2016) studied Oreocallis grandiflora and found that robbed flowers received fewer visits than undamaged ones; however, this did not result in a decrease in the overall seed set. This outcome appears to be influenced by other factors such as the plant’s breeding system and the pollinator community. In contrast, Anthyllis vulneraria subsp. vulgaris experiences an increase in fruit set despite nectar robbing, likely due to the behavior of its primary pollinators, bumblebees, which access both reproductive structures while foraging for nectar from robbed flowers (Navarro, 2000).

The temperate rainforest of southern South America is home to numerous plant species with long corollas that produce abundant nectar. The hummingbird Sephanoides sephaniodes primarily pollinates these plants and they are also visited by the native bumblebee Bombus dahlbomii, which are the only two native species known to consistently visit and pollinate these long-corolla flowers in this ecosystem (Aizen, Vázquez & Smith-Ramírez, 2002). One such plant is Fuchsia magellanica, a native shrub that experiences primary nectar robbing by the native bird Phrygilus patagonicus and by the introduced European bumblebee Bombus terrestris (Traveset, Willson & Sabag, 1998; Valdivia, Carroza & Orellana, 2016). The antagonistic relationship between F. magellanica and P. patagonicus likely began 16 to 17 million years ago, when this shrub started to be pollinated by S. sephaniodes (Abrahamczyk & Renner, 2015). In contrast, the nectar robbing by B. terrestris is more recent, emerging only in the late 1990s when this bumblebee was introduced for crop pollination (Valdivia, Carroza & Orellana, 2016; Smith-Ramírez et al., 2018; Vieli et al., 2021).

Fuchsia magellanica is a widely distributed shrub native to the temperate rainforests and coastal scrublands of southern South America, extending from central Chile and Argentina to Tierra del Fuego (Hoffmann, 1997). It plays a key ecological role as one of the most abundant and reliable nectar sources for native pollinators during the flowering season (Aizen, Vázquez & Smith-Ramírez, 2002). The species has a mixed mating system, combining self-compatibility with a strong dependence on pollinators for optimal seed production, making it particularly sensitive to changes in pollinator communities (Riveros, Humaña & Arroyo, 1996). The native forest matrix where F. magellanica thrives is a biodiverse temperate ecosystem characterized by high endemism and complex plant–pollinator networks (Aizen, Vázquez & Smith-Ramírez, 2002; Guimarães et al., 2025). The introduction of the European bumblebee Bombus terrestris in Chile in the late 1990s for crop pollination has led to its rapid spread and establishment throughout much of southern South America (Smith-Ramírez et al., 2018; Fontúrbel, Murúa & Vieli, 2021). This species is now recognized as a global invasive pollinator that not only competes with native pollinators such as B. dahlbomii and S. sephaniodes but also disrupts plant–pollinator dynamics, altering visitation rates, foraging behavior, and, in some cases, reproductive success of native plants (Smith-Ramírez et al., 2023; Guimarães et al., 2025). Its ongoing expansion, facilitated by human-mediated introductions and its ecological flexibility, represents a major challenge for the conservation of native pollination networks in these temperate ecosystems, making it crucial to understand its ecological impact in the unique context of South American temperate forests (Guimarães et al., 2025).

Previous work has shown that the invasive bumblebee B. terrestris exhibits broad geographic variation in nectar-robbing behavior on F. magellanica across southern Chile, forming a geographic mosaic of interactions (Valdivia, Carroza & Orellana, 2016). This variation is primarily driven by two factors: (i) phenotypic mismatches between floral traits, particularly corolla length, and the morphology of bumblebee mouthparts; and (ii) differences in the local abundance of B. terrestris populations (Valdivia, Carroza & Orellana, 2016; Rosenberger et al., 2021). In populations where flowers have longer corollas and bumblebees have shorter tongues, the likelihood of nectar robbing increases substantially (Valdivia, Carroza & Orellana, 2016). Similarly, areas with high population densities of B. terrestris experience more intense nectar robbing (Rosenberger et al., 2021). Together, these factors create a spatially structured mosaic in which some populations endure consistently higher rates of nectar robbing, while others experience little or none (Valdivia, Carroza & Orellana, 2016). Although these geographic and phenotypic patterns are well documented, the reproductive consequences of such variation, particularly in terms of seed production, remain unknown. Understanding these outcomes is crucial to evaluate the ecological and evolutionary implications of invasive pollinators in native plant–pollinator networks.

The primary objective of this study was to evaluate the impact of nectar robbing by the invasive bumblebee Bombus terrestris on nectar production, the foraging behavior of key pollinators, and the reproductive success of Fuchsia magellanica. Specifically, the study addressed the following questions: (i) How do the rates of nectar production and the standing crop of nectar change as a result of nectar robbing by B. terrestris? (ii) How does the foraging behavior of flower visitors, including both native and exotic species, shift in response to primary nectar robbing by B. terrestris? and (iii) Do these changes in pollinator behavior ultimately reduce the female reproductive success of F. magellanica? To answer these questions, we quantified the percentage of flowers pierced per plant, nectar production rates and standing crop, the frequencies of both pollinating and robbing visits, and the seed set of plants across 24 populations of F. magellanica in southern Chile.

Materials & Methods

Sites and species under study

Fieldwork was conducted between January and March in both 2012 and 2013 in the temperate forests of southern Chile. All work complied with Chilean regulations governing research on native flora and fauna, which do not require permits for observational studies without specimen collection. Institutional approval was obtained through the Universidad de Los Lagos. This region covers an area of 67,012 km2, located within the Los Ríos and Los Lagos regions, and features an altitudinal gradient exceeding 1,300 m, shaped by the Andean and coastal mountain ranges (Fig. 1A). It is a diverse area with a variety of habitat types, including native evergreen and deciduous forests, mixed with forestry plantations of Pinus species and Eucalyptus species, grasslands for livestock, and urban areas (Luebert & Pliscoff, 2023). The forests, composed of native species, host numerous plants with long corolla tubes primarily pollinated by the hummingbird S. sephaniodes, including Embothrium coccineum, Tristerix corymbosus, and Lapageria rosea. Among these plants is the shrub F. magellanica.

Figure 1 Study area and nectar robbing by Bombus terrestris in Fuchsia magellanica.

(A) Study area in the Los Ríos and Los Lagos regions of southern Chile, showing the 24 populations of F. magellanica included in the study. Populations are numbered in alphabetical order; red circles indicate populations where primary nectar robbing by B. terrestris was recorded, while green circles indicate populations without records of nectar robbing. For details on the mean percentage (±1 SE) of robbed flowers per population, see Appendix 1. (B) Example of a B. terrestris individual performing nectar robbing on a flower of F. magellanica. Holes pierced by B. terrestris can later be exploited by both native and exotic floral visitors, leading to secondary nectar robbing (Photo: Alberto Gantz). The base map was generated from a screenshot of Macrostrat (https://macrostrat.org/map/#x=-72.749&y=-41.52&z=6.5) and edited in PowerPoint. No copyrighted or third-party graphics were used.

In the study area, 24 sites were selected at which at least 20 adult plants of F. magellanica were found (Fig. 1A; Appendix 1). Site selection was informed by previous observations and published data documenting varying levels of primary nectar robbing by the non-native bumblebee B. terrestris (Fig. 1B; Valdivia, Carroza & Orellana, 2016). Fuchsia magellanica is a long-lived shrub with a mixed mating system: it is partially self-compatible, but fruit and seed production rely largely on pollinator activity (Riveros, Humaña & Arroyo, 1996). The plant produces pendulous tubular flowers approximately 5–12 mm in diameter and 40–50 mm long, with sepals measuring 12–25  × 3–8 mm and petals 7–20  × 4–7 mm (Hoffmann, 1997; Riveros, Humaña & Arroyo, 1996; CE Valdivia, pers. obs., 2012). Individual flowers last several days, and the plant can bloom from spring through autumn (Valdivia, Carroza & Orellana, 2016; CE Valdivia, pers. obs., 2012). The plant produces elongated berries containing numerous small seeds, with fruiting typically occurring between November and May (Armesto et al., 1987; Hoffmann, 1997; Morales-Paredes, Orellana & Valdivia, 2018). In addition, the species interacts with several frugivores, including birds and lizards (Morales-Paredes, Orellana & Valdivia, 2018).

Primary nectar robbing and floral reward

To analyze the impact of primary nectar robbing on nectar production rates and the standing crop of nectar of F. magellanica, we selected twenty plants per site, totaling 480 plants. Initially, we evaluated the effect of primary nectar robbing by comparing pierced flowers, damaged by B. terrestris, to undamaged flowers. We selected ten flowers per plant (n = 10 flowers × 20 plants ×24 populations = 4,800 flowers), which we inspected visually to identify floral damage. Pierced flowers were recognized by the presence of a small hole or slit at the base or side of the corolla tube, typically caused by Bombus terrestris during nectar robbing, whereas undamaged flowers showed no evidence of such perforations. All mature flowers were classified as either pierced or undamaged, noting the unequal representation of the two categories. To avoid sampling bias, site, plant, branch, and flower selection were all conducted randomly. On each plant, branches were selected at random, and mature flowers were chosen haphazardly from those available, without preference for size or condition. This ensured that sampling reflected the natural variation present in each population.

We estimated the nectar secretion rate in intact and undamaged flowers by bagging them (Corbet, 2003). This approach allowed us to eliminate any potential gain or loss of water and solutes from animal interactions. We used bridal-veil bags to bag five flowers from each of 20 plants across 24 populations, resulting in a total of 2,400 flowers. To prevent contact with pollinators or nectar robbers, these flowers were left bagged for 24 h. After this period, we carefully removed the flowers from the plants to extract the nectar using a capillary tube (75 mm length, 1.5 mm diameter). The amount of nectar obtained was then measured with an electronic digital caliper (Corbet, 2003).

We estimated the standing crop of nectar due to its high variability across time and space. Standing crop was defined as the average amount of nectar available per flower at a given time. This was measured directly from exposed flowers during the observation period and does not refer to the total nectar of a whole plant or population. This measurement is better understood as a reflection of the current interactions between a population of flowers and a population of nectar foragers (Corbet, 2003). To evaluate the impact of primary nectar robbing on the standing crop of nectar, specifically in flowers exposed to floral visitors, we randomly selected ten mature flowers from each of 20 plants during the peak flowering season (resulting in a total of 10 flowers × 20 plants × 24 sites = 4,800 flowers). Each flower was visually inspected and classified as either pierced or undamaged, leading to an unequal representation of the two categories. We carefully cut the flowers from their branches and immediately extracted the nectar using capillary tubes. The amount of nectar was then measured with an electronic digital caliper. Nectar measurements were recorded and analyzed separately for pierced and unpierced flowers. Because F. magellanica shrubs often bear dozens of open flowers, we did not census all flowers per plant. Instead, we estimated standing crop on a per-flower basis, averaging values within each category at the plant level to allow comparisons between damaged and undamaged flowers across populations. In populations where pierced flowers were present but completely depleted of nectar, these values were recorded as 0 µl. In populations where no pierced flowers were present, no measurements could be taken for that condition.

To evaluate the effects of primary nectar robbing on nectar production rates and the standing crop of nectar, we fitted two generalized linear mixed models. In both models, we considered the percentage of pierced flowers per plant, the identity of the population, the status of the flower (pierced vs. undamaged), and all two-way interactions as fixed effects. Computations were conducted using the Statistica software package v. 10.0 (StatSoft Inc., Tulsa, OK, USA).

Primary nectar robbing and foraging behavior of pollinators

To evaluate the effects of primary nectar robbing by B. terrestris on the rates of pollinating and robbing visits to F. magellanica flowers, we used the same 20 plants at each site. For these plants, five flowering branches were selected haphazardly, without applying any systematic rule or preference for their position, size, or flower number, and then marked. In total, we tagged and observed an average of 15.1 ± 0.1 (mean ± 1 SE) flowers per plant for 10 min in the morning (between 8:00 AM and 12:00 PM), conducting this observation only once around the peak of the flowering season. All observations were performed on sunny or partially cloudy days to avoid any potential bias due to weather conditions. During each observation period (one period per individual plant), we recorded the identity of floral visitors, and the type of foraging behavior exhibited (i.e., pollinating vs. robbing visits). A pollinating visit was defined as a feeding behavior in which an animal landed or hovered over F. magellanica while contacting the reproductive structures (i.e., anthers and/or stigma) of the flowers. In contrast, a robbing visit was characterized as a feeding behavior where an animal landed or hovered over F. magellanica piercing a previously undamaged flower to forage (i.e., primary nectar robbery) (Inouye, 1980). Alternatively, it was considered secondary robbing if the animal foraged from flowers that had already been pierced by B. terrestris (Inouye, 1980).

Plant and insect identifications followed regional floras and faunal guides. Insect identifications were also confirmed by specialists familiar with the fauna of southern Chile. Sephanoides sephaniodes, B. terrestris, and B. dahlbomii are readily distinguishable in the field, with B. dahlbomii being unmistakable due to its diagnostic orange and black coloration. All observers had previous training in species identification and behavioral recording. Observations were conducted at approximately 3–4 m from focal plants to minimize disturbance. No behavioral changes in hummingbirds attributable to observer presence were detected during preliminary trials or the study itself. Because this was an observational field study on naturally occurring plants and flower visitors, no additional permits were required beyond institutional approvals for fieldwork (project number: 11110230).

To assess the effects of primary nectar robbing on the foraging rates of floral visitors, we employed a generalized linear mixed model. In this model, we incorporated the percentage of pierced flowers per plant, the identity of the population, the type of foraging behavior (i.e., pollinating visits vs. robbing visits), and all two-way interactions as fixed effects. The computations were conducted using the Statistica software package v. 10.0 (StatSoft Inc., Tulsa, OK, USA).

Primary nectar robbing and seed set

To evaluate the effects of nectar robbing on seed production, we selected 15 flowers per plant from each of the 20 plants sampled at each population. These flowers were randomly assigned to three treatments, with five flowers per treatment (15 flowers per plant in total). (i) Exclusion of nectar robbers and pollinators (-Rob -Pol): All floral visitors, including both pollinators and nectar robbers, were excluded by bagging flower buds with bridal-veil bags until they produced mature fruits (approximately 30–45 days later). (ii) Exclusion of nectar robbers with pollinator presence (-Rob +Pol): Only nectar robbers were excluded from the flowers by covering the nectary chamber with masking tape at the bud stage and keeping it in place until the flowers produced mature fruits. (iii) Presence of nectar robbers and pollinators (+Rob +Pol): All flowers were tagged inconspicuously until fruit maturity.

At the end of the experiments, we collected the fruits produced from flowers subjected to these trials and stored them in a 90% alcohol solution until seed set assessment was conducted under laboratory conditions. Subsequently, seeds from each fruit were released and counted using a stereoscopic magnifier. The measures of seed production for each experimental condition were averaged for each plant, resulting in a total of 480 experimental plants across the three conditions. However, some plants were lost during the assessment process. Despite this, the protocol enabled us to evaluate the quantitative effects (i.e., the number of seeds per pollination trial) of nectar robbing by B. terrestris on the seed set of F. magellanica.

To analyze the effects of primary nectar robbing on seed set, we fitted a generalized linear mixed model. This model included the percentage of pierced flowers per plant, population identity, pollination trial (i.e., -Pol -Rob vs. +Pol -Rob vs. +Pol +Rob), and all two-way interactions as fixed effects. The computations were carried out using the Statistica software package (version 10.0, StatSoft Inc., Tulsa, OK, USA).

Results

Primary nectar robbing and floral reward

Both the nectar production rate and the standing crop of nectar varied significantly based on the status of the flowers (i.e., undamaged vs. pierced) and among different populations (see Tables 1A, 1B; Appendix 2). Pierced flowers produced nectar at rates and had standing crops that were 4 times and 2.9 times lower than those of undamaged flowers, respectively (see Fig. 2).

However, the percentage of pierced flowers within a population did not significantly affect the nectar production rate for either undamaged or pierced flowers (Table 1A). The regression analysis showed the following results: (a) For undamaged flowers, β ± SE = −0.015 ± 0.071, t = −0.205, P = 0.84; (b) For pierced flowers, β ± SE = 0.006 ± 0.035, t = 0.176, P = 0.86. Similarly, the percentage of pierced flowers did not impact the standing crop of nectar for undamaged or pierced flowers (Table 1B). The results were as follows: Undamaged flowers: β ± SE = −0.052 ± 0.033, t = −1.544, P = 0.13; Pierced flowers: β ± SE = −0.017 ± 0.022, t = −0.752, P = 0.46. Overall, these results indicate that although nectar levels vary across populations and flower status, the proportion of pierced flowers per population does not significantly explain this variation.

Table 1 Results of linear mixed models testing the effects of nectar robbing on nectar dynamics in Fuchsia magellanica.

Effects of the percentage of pierced flowers, population identity, and flower status (pierced vs. undamaged) on (a) nectar production rate and (b) standing crop of nectar. Significant effects (P < 0.05) are shown in bold.

Source	df	MS	F	p-level	
(a) Nectar production rate:					
Percentage of pierced flowers	1	0.538	0.013	0.910	
Population	8	256.616	6.176	<0.001	
Status	1	269.859	4.753	0.038	
Status*Percentage of pierced flowers	1	3.360	0.059	0.810	
Status*Population	8	176.239	3.104	0.012	
Error	28	56.776			
(b) Standing crop of nectar:					
Percentage of pierced flowers	1	36.306	2.940	0.097	
Population	8	48.939	3.963	0.003	
Status	1	65.379	5.154	0.031	
Status*Percentage of pierced flowers	1	9.563	0.754	0.393	
Status*Population	8	4.935	0.389	0.917	
Error	28	12.686	 	 	

Figure 2 Effects of primary nectar robbing on nectar production and standing crop in Fuchsia magellanica.

(A) Nectar production rate and (B) standing crop of nectar in undamaged and pierced flowers. Pierced flowers represent primary nectar robbing by the invasive bumblebee Bombus terrestris. Different letters indicate significant differences between treatments (P < 0.05; Tukey HSD tests following mixed linear models).

Primary nectar robbing and foraging behavior of pollinators

Fuchsia magellanica flowers were visited by eleven different species: three of these were birds and eight were insects. These visitors varied in their visitation and foraging behaviors, with some acting as mutualists (i.e., pollinators) and others as antagonists (i.e., primary nectar robbers), as shown in Table 2. Notably, four species were the most frequent visitors to the flowers: the native hummingbird S. sephaniodes (Trochilidae), the native bumblebee B. dahlbomii (Apidae), the exotic bumblebee B. terrestris (Apidae), and the exotic honeybee Apis mellifera (Apidae).

Table 2 Mutualist and antagonist floral visitors of Fuchsia magellanica.

Floral visitors were categorized according to their foraging behavior following Inouye (1980): primary nectar robbers (PNRob), secondary nectar robbers (SNRob), nectar thieves (NT), and pollinators (Pol). The table also indicates the floral rewards obtained (nectar and/or pollen), the origin of the visitor (native or exotic), and whether their activity caused damage to the reproductive structures (ovary or stamens) or to sepals and petals.

Flower visitors	Family	Origin	Reward	Foraging
behavior	Damage to	
					Ovary or stamens	Sepals or petals	
(a) Birds:							
Elaenia albiceps	Tyrannidae	Native	Nectar	SNRob	No	No	
Phrygilus patagonicus	Thraupidae	Native	Nectar	PNRob	Yes	Yes	
Sephanoides sephaniodes	Trochilidae	Native	Nectar	Pol	No	No	
(b) Insects:							
Apis mellifera	Apidae	Exotic	Pollen/Nectar	Pol/SNRob	No	No	
Argopteron aureipennis	Hesperiidae	Native	Nectar	Pol/NT	No	No	
Bombus dahlbomii	Apidae	Native	Nectar	Pol/SNRob	No	No	
Bombus terrestris	Apidae	Exotic	Nectar	PNRob/Pol	No	Yes	
Diphaglossa gayi	Colletidae	Native	Nectar	Pol	No	No	
Eroessa chilensis	Pieridae	Native	Nectar	Pol	No	No	
Lasia nigritarsis	Acroceridae	Native	Nectar	NT	No	No	
Vespula germanica	Vespidae	Exotic	Nectar	SNRob	No	No	

The hummingbird S. sephaniodes exclusively engaged in pollinating visits, while the insect species exhibited mixed foraging behaviors that included both pollinating and robbing visits. During pollinating visits, all flower visitors contacted the reproductive structures of the flowers (anthers and/or stigma). When they conducted robbing visits, some of them also touched these reproductive structures. Pollinating visits by S. sephaniodes, B. dahlbomii, and B. terrestris occurred as the animals extracted nectar through the flower’s entrance. However, A. mellifera’s pollinating visits were characterized by collecting pollen. All robbing visits were carried out by animals sucking nectar through holes that B. terrestris had previously bitten into the walls of the nectary chamber. Thus, every robbing visit was a case of secondary nectar robbing.

The frequency of visits by floral visitors varied significantly based on foraging behavior (i.e., pollinating or robbing visits) and among different populations (see Table 3A; Appendix 3). Robbing visits were observed more frequently than pollinating visits, accounting for 53.1% and 46.9% of all visits, respectively. Pollinating visits were primarily carried out by four species: B. dahlbomii (mean ± SE: 15.8  ± 19.4% of visits), A. mellifera (14.2 ±20.2%), S. sephaniodes (11.9 ± 18.9%), and B. terrestris (5.0 ± 11.1%). In contrast, robbing visits were mainly conducted by B. terrestris (49.9 ±0.1%), followed by B. dahlbomii (2.0 ± 0.0%) and A. mellifera (1.1 ± 0.0%) (see Fig. 3). It is important to note that not all of these species were present in all populations, which contributed to the observed variation in visitation frequencies.

Table 3 Linear mixed models for visitation rates and seed set in Fuchsia magellanica.

(a) Model testing the effects of the percentage of pierced flowers, population identity, and foraging behavior (pollinating vs. robbing visits) on the rate of visits to flowers. (b) Model testing the effects of the percentage of pierced flowers, population identity, and pollination treatment (–Rob–Pol, –Rob+Pol, +Rob+Pol; where Pol, pollinators and Rob, nectar robbers) on seed set.

Source	df	MS	F	p-level	
(a) Rate of visits to flowers:					
Percentage of pierced flowers	1	0.027	2.264	0.133	
Population	23	0.049	4.171	<0.001	
Behavior	1	0.096	8.205	0.004	
Behavior*Percentage of pierced flowers	1	0.195	16.546	<0.001	
Behavior*Population	23	0.045	3.795	<0.001	
Error	455	0.012			
(b) Seed set:					
Percentage of pierced flowers	1	691.988	0.278	0.598	
Population	23	20,377.107	8.186	<0.001	
Pollination trial	2	36,182.407	15.738	<0.001	
Pollination trial*Percentage of pierced flowers	2	1,300.001	0.565	0.568	
Pollination trial*Population	46	8,743.937	3.803	<0.001	
Error	808	2,299.052	 	 	

Figure 3 Pollinating and robbing visits by the main floral visitors of Fuchsia magellanica.

Mean (±SE) number of visits per flower per 10 min, separated by foraging behavior (pollinating vs. robbing). Panels show (A) Apis mellifera, (B) Bombus dahlbomii, (C) Bombus terrestris, and (D) Sephanoides sephaniodes. For B. terrestris, only secondary nectar robbing events are plotted. Although primary nectar robbing was occasionally observed in the field, it was extremely rare and did not occur during the standardized observation periods used to calculate visit rates.

The percentage of pierced flowers did not influence the overall rate of visits to flowers when pollinating visits were analyzed together with robbing visits (Table 3A). However, regression analysis showed significant negative effects of the percentage of pierced flowers on the rate of pollinating visits (Pollinating visits: β ± SE = −0.0006 ±0.0003, t = −2.087, P = 0.04; see Fig. 4). Additionally, a significant positive effect of the percentage of pierced flowers on robbing visits was observed (β ± SE = 0.001 ±0.0003, t = 3.529, P < 0.001; see Fig. 4).

Figure 4 Relationship between primary nectar robbing and flower visitation rates.

Predicted rates of pollinating and robbing visits (number of visits per flower per 10 min) as a function of the percentage of pierced flowers per plant. Pollinating visits were correlated negatively with the proportion of pierced flowers, while robbing visits were positively correlated. Solid lines show model predictions and dashed lines the 95% confidence intervals from generalized linear mixed models. Negative values on the Y-axis are an artifact of the model and should be interpreted as visitation rates approaching zero.

Primary nectar robbing and seed set

The seed set varied significantly based on the pollination trial, specifically the presence or absence of pollinators and/or nectar robbers, as well as among different populations (Table 3B; Appendix 4). Flowers that were excluded from both nectar robbers and pollinators (-Rob-Pol) produced 5.4 to 4.6 times fewer seeds compared to flowers that had nectar robbers excluded but were exposed to pollinators (-Rob+Pol), and those exposed to both nectar robbers and pollinators (+Rob+Pol), respectively (Fig. 5). There were no significant differences in seed set between the pollination trials -Rob+Pol and +Rob+Pol (Fig. 5).

Figure 5 Seed set under pollination and nectar robbing treatments.

Mean (±SE) number of seeds per flower of Fuchsia magellanica subjected to three experimental treatments: flowers excluded from both pollinators and nectar robbers (−Pol −Rob), flowers accessible to pollinators but protected from nectar robbers (+Pol −Rob), and flowers accessible to both pollinators and nectar robbers (+Pol +Rob). Different letters above bars indicate significant differences (P < 0.05) according to Tukey HSD tests following a linear mixed model.

The percentage of pierced flowers did not influence seed set when the three levels of the pollination trial (i.e., -Rob-Pol, -Rob+Pol, and +Rob+Pol) were analyzed together (see Table 3B). As a result, regression analyses between the percentage of pierced flowers and the -Rob+Pol and +Rob+Pol treatments were not significant (-Rob+Pol: β ± SE = −0.040 ± 0.179, t = −0.225, P = 0.82; +Rob+Pol: β ± SE = 0.068 ±0.232, t = 0.292, P = 0.77). In contrast, a regression analysis between the percentage of pierced flowers and the -Rob-Pol treatment was significant and negative (β ± SE = −0.195 ± 0.095, t = −2.055, P = 0.04).

Discussion

Nectar robbing by the invasive bumblebee B. terrestris significantly reduced the amount of nectar available in the flowers of F. magellanica. This decrease in nectar quantity, along with the presence of holes in the flowers, altered the foraging behavior of flower visitors. As a result, the rate of pollinating visits decreased, while the rate of secondary nectar robbing increased. Despite these changes in the quantity and quality of flower visits, shifting from primarily pollinating visits to mostly robbing visits, there was no decline in the seed set of the plants, even though they have a moderate to high dependency on pollinators.

The small effect sizes we detected in the relationship between the proportion of pierced flowers and visitation patterns may partly reflect the spatial scale of our analyses, which were conducted at the population level. Pollinator displacement is often more evident at the patch scale, where local foraging decisions can be detected more clearly (Maloof, 2001; Irwin & Brody, 1998). Moreover, effective displacement would require the presence of alternative and accessible floral resources in the surrounding landscape. In our study sites, the availability of such resources was likely limited or unevenly distributed, potentially reducing opportunities for observable displacement and thus attenuating the strength of the effect detected (Irwin et al., 2010).

The dynamics of nectar production in flowers are defined by temporary patterns of secretion, cessation, and reabsorption (Nicolson, 2007). Like other flower traits, these aspects are influenced by genetic and environmental factors, which can lead to significant variations among different populations (Nicolson, 2007). In the case of F. magellanica, the interaction between these factors likely explains the differences in daily nectar production rates of undamaged flowers across populations. Additionally, some populations experience geographical variation in nectar production rates due to nectar robbing by B. terrestris. Pierced flowers tend to have lower nectar production rates compared to undamaged flowers. However, the daily nectar production rates of both undamaged and pierced flowers were not influenced by the percentage of pierced flowers on the plant. This indicates that the nectar yield of any individual flower is not affected by the ratio of robbed flowers on the same plant.

There appear to be three main factors contributing to the reduction in daily nectar production rates in F. magellanica due to primary nectar robbing. First, decreased nectar production may result from tissue damage, as reported in other nectar-robbing systems (Irwin & Brody, 1998; Parachnowitsch, Manson & Sletvold, 2019). Second, the higher rate of nectar evaporation may occur through holes created by B. terrestris, especially in plants exposed to greater solar radiation or windy conditions (Nicolson, 2007). Finally, field observations suggest that nectar may spill out of pierced flowers, particularly during strong winds or heavy rainfall, leading to a net reduction in available nectar. Further experimental work would be needed to determine the relative importance of these factors.

The rates of nectar production play a crucial role in determining the overall amount of nectar available, which can also be influenced by primary nectar robbing (Nicolson, 2007). Specifically, flowers of F. magellanica that have been pierced exhibit a lower standing crop of nectar compared to undamaged flowers. This reduction in nectar availability in F. magellanica has been noted in other plants that are pollinated by hummingbirds, such as Justicia aurea, Aphelandra golfodulcensis, Pavonia dasypetala, and Ipomopsis aggregata (McDade & Kinsman, 1980; Irwin & Brody, 1998).

The standing crop of nectar is a crucial factor for plant reproduction because it relies on a mutually dependent relationship with animal visits. The foraging behavior of visiting animals is influenced by the standing crop, which is also affected by animal activity (Nicolson, 2007). This interdependent relationship is further shaped by the animals’ ability to perceive their environment (Blarer, Keasar & Shmida, 2002; Armbruster, Antonsen & Pélabon, 2005; Fenster et al., 2006). Floral visitors can directly choose among flowers by evaluating the presence and quantity of rewards, provided they can perceive these benefits. However, if they are unable to directly assess the reward, they must rely on indirect clues related to the rewards when selecting flowers (Blarer, Keasar & Shmida, 2002; Armbruster, Antonsen & Pélabon, 2005; Fenster et al., 2006). This latter scenario is common in plants pollinated by hummingbirds, where nectar is hidden within nectary chambers. In the case of F. magellanica, animals can indirectly evaluate the amount of nectar available in undamaged or pierced flowers by observing the presence or absence of holes created by B. terrestris.

If animals are unable to associate the advertising cues (such as holes) with the rewards (like nectar), it may lead to a greater emphasis on pollinators assessing rewards directly (Irwin, 2000). To maximize their net energy intake while minimizing total foraging time, pollinators are expected to avoid plants that have been robbed of nectar (Irwin, 2000). As a result, nectar robbing will select for pollinators capable of learning to avoid plants and flowers depleted of nectar (González-Gómez & Vásquez, 2006; González-Gómez et al., 2014; González-Gómez, Bozinovic & Vásquez, 2011). This phenomenon appears to be true for S. sephaniodes, which is the only pollinator that reduces the rate of visits to robbed plants. Compared to other pollinators, S. sephaniodes has higher energy demands and presumably superior learning and memory skills, which likely drive this foraging behavior (Fernández, López-Callejas & Bozinovic, 2002; González-Gómez & Vásquez, 2006; González-Gómez et al., 2014; González-Gómez, Bozinovic & Vásquez, 2011).

A decrease in hummingbird visits is associated with an increase in secondary nectar robbing behavior by B. terrestris, which occurs because of primary nectar robbing. This behavior, exhibited by the invasive B. terrestris, is a socially transmitted foraging strategy that spreads rapidly within populations and creates positive feedback loops (Leadbeater & Chittka, 2008; Sherry, 2008). A similar phenomenon appears to occur with the native bumblebee B. dahlbomii, which also shows an increase in the rate of robbing visits without a corresponding decrease in pollination visits. The cheating behavior observed in the native bumblebee may originate from observing B. terrestris engage in secondary nectar robbing. This behavior could be sustained over time through feedback among individuals of both species, driven by both inter- and intraspecific social transmission. Such contagious behavior between and within species has been documented in other bumblebee species, such as between Bombus wurflenii and Bombus lucorum (Goulson et al., 2013), but confirmation for B. dahlbomii and B. terrestris is still needed.

Seed production in F. magellanica is heavily reliant on pollinators, yet our results provide no significant evidence that nectar robbing reduces seed set for two main reasons. First, when B. terrestris engages in primary nectar robbing, it does not damage the flowers’ fertile whorls (i.e., stamens and style). As a result, there is no direct impact on the seed set. Second, while primary nectar robbing does increase instances of secondary nectar robbing, it simultaneously decreases the frequency of pollinator visits. Despite this, F. magellanica still experiences a gradual decline in the total number of pollinating visits because the pollination behavior of B. dahlbomii remains largely unaffected by primary nectar robbing. Therefore, there is no indirect reduction in seed set mediated by pollinators. This ecological scenario, in which seed production in plants remains unaffected by nectar robbers, has been observed in numerous other species (Maloof & Inouye, 2000; Irwin et al., 2010). Therefore, our results are consistent with the possibility that pollinators are not currently a limiting factor for seed set in F. magellanica. Although pollination dynamics were altered by nectar robbing, seed set did not decline significantly, suggesting that the pollination service provided by the native assemblage remains sufficient to ensure reproduction. This pattern aligns with other systems where plants with generalized or flexible pollination strategies maintain seed production despite disturbances to plant–pollinator interactions (Bennett et al., 2020).

The large variation in seed set observed across populations, even under similar treatments, may reflect the influence of multiple local-scale factors (Knight et al., 2005). Microclimatic conditions such as rainfall, solar radiation, and temperature can affect both nectar production and pollinator activity (Nicolson, 2007; Lawson & Rands, 2019). Likewise, soil fertility and nutrient availability may influence plant reproductive output (Carvalheiro et al., 2021). Differences in the composition and relative abundance of local pollinator assemblages may further contribute to this variability, particularly where the invasive B. terrestris co-occurs with native pollinators (Ryniewicz et al., 2022). These interacting factors likely explain the observed heterogeneity in reproductive success among populations.

Finally, although our experimental design allowed us to isolate the effects of nectar robbing on reproductive success, some methodological caveats should be considered. The use of masking tape to restrict access in the −Rob+Pol treatment may have had subtle effects on pollinator behavior or fruit set, although we did not observe any behavioral changes in the field that would suggest such an effect. Additionally, while the treatments effectively controlled access by robbers, they did not manipulate pollinator visitation rates directly. Thus, we cannot completely rule out the possibility that differences in visitation rates between the −Rob+Pol and +Rob+Pol treatments could have influenced seed set. Future studies could incorporate additional controls or direct measurements of visitation rates to better evaluate these potential effects.

Conclusions

Mutualisms between plants and pollinators can be threatened by nectar robbers, which consume nectar without providing the important service of pollination. In the temperate forests of Chile, the shrub F. magellanica is mainly pollinated by the hummingbird S. sephaniodes and the native bumblebee B. dahlbomii. However, some populations are also affected by nectar robbing from the invasive bumblebee B. terrestris. The primary objective of this study was to evaluate the impact of nectar robbing on nectar production, the foraging behavior of key pollinators, and the reproductive success of F. magellanica. Specifically, the study aimed to answer the following questions: (i) How do nectar production rates and the standing crop of nectar change as a result of nectar robbing? (ii) How does the foraging behavior of flower visitors change due to nectar robbing? and (iii) Does the change in pollinator behavior lead to a reduction in the female reproductive success of F. magellanica?

The results indicated that nectar robbing reduced nectar production rates by a factor of 4 and decreased the standing crop of nectar by a factor of 2.9. As a result of an increased percentage of flowers pierced by B. terrestris, the hummingbird S. sephaniodes reduced its pollination visits, while B. terrestris increased its nectar robbing visits. Despite these changes in pollinator behavior, there was no significant effect on seed set. The primary nectar robbing by B. terrestris on F. magellanica flowers does not appear to affect seed production as initially expected. This indicates that, despite its significant influence on pollinator behavior, this invasive bumblebee does not seem to threaten the persistence of the plant population. However, further evaluation is necessary to determine whether B. terrestris poses ecological and evolutionary risks to other hummingbird-pollinated plants in the temperate rainforests of South America.

Supplemental Information

Supplemental Information 1 Raw data

Dataset containing field measurements of flower visitation, nectar production, nectar standing crop, and seed set of Fuchsia magellanica across 24 populations in southern Chile. Data correspond to the analyses presented in the main text and supplementary appendices.

Supplemental Information 2 Primary nectar robbing of Fuchsia magellanica across study populations

Mean percentage (±1 SE) of Fuchsia magellanica flowers pierced by Bombus terrestris (primary nectar robbing) in 24 populations across the Los Ríos and Los Lagos regions of southern Chile. Populations are listed in alphabetical order and correspond to the numbering used in Figure 1

Supplemental Information 3 Nectar production and standing crop in pierced and undamaged flowers of Fuchsia magellanica

Mean nectar production rate (μl/flower/day) and standing crop of nectar (μl/flower) in pierced and undamaged flowers of Fuchsia magellanica across 24 populations in southern Chile. Values are means ± 1 SE. “0” indicates flowers measured but with no nectar present; “nr” indicates that no pierced flowers were present in the population, so no measurements could be taken

Supplemental Information 4 Foraging behavior and visitation rates of floral visitors to Fuchsia magellanica

Mean (±1 SE) rates of pollinating and robbing visits (number of visits per flower per 10 min) and mean visitation frequencies (%) by the hummingbird Sephanoides sephaniodes , the native bumblebee Bombus dahlbomii , the exotic bumblebee B. terrestris , and the exotic honeybee Apis mellifera across 24 populations of Fuchsia magellanica in southern Chile

Supplemental Information 5 Seed set of Fuchsia magellanica under different experimental conditions

Mean seed set (number of seeds per flower ± 1 SE) of Fuchsia magellanica across 24 populations in southern Chile. Experimental conditions included: absence of pollinators and robbers (−Pol −Rob), presence of pollinators without robbers (+Pol −Rob), and presence of both pollinators and robbers (+Pol +Rob)

We are grateful to José P. Carroza, Alberto Gantz, Soraya Sade, and Carlos Morales for their valuable comments and assistance during fieldwork. All experiments complied with the current laws of Chile, where the research was conducted. The English of the manuscript was improved using AI-based tools (Grammarly and ChatGPT, OpenAI).

Additional Information and Declarations

Competing Interests

Author Contributions

Field Study Permissions

Data Availability

The authors declare there are no competing interests.

Carlos E. Valdivia conceived and designed the experiments, performed the experiments, analyzed the data, prepared figures and/or tables, authored or reviewed drafts of the article, conceptualization, Methodology, Validation, Formal analysis, Research, Resources, Data curation, Writing—Original draft preparation, Writing—review and editing, Visualization, Supervision, Project administration, Funding acquisition, and approved the final draft.

José I. Orellana conceived and designed the experiments, performed the experiments, analyzed the data, prepared figures and/or tables, authored or reviewed drafts of the article, conceptualization, Methodology, Validation, Formal analysis, Research, Resources, Data curation, Writing—Original draft preparation, Writing—review and editing, Visualization, and approved the final draft.

Maureen Murúa conceived and designed the experiments, performed the experiments, analyzed the data, prepared figures and/or tables, authored or reviewed drafts of the article, conceptualization, Methodology, Validation, Formal analysis, Research, Resources, Data curation, Writing—Original draft preparation, Writing—review and editing, Visualization, and approved the final draft.

The following information was supplied relating to field study approvals (i.e., approving body and any reference numbers):

Field experiments were approved by the Universidad de Los Lagos and Fondecyt (project number: 11110230).

The following information was supplied regarding data availability:

The raw measurements are available in the Supplementary File.

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
