# Peer review of "Nectar robbing by the invasive bumblebee Bombus terrestris (Apidae) changes the behavior of native flower visitors of Fuchsia magellanica Lam. (Onagraceae) but not seed set"

_PeerJ, doi:10.7717/peerj.20253_

## Round 0.1 · original submission · Major Revisions

Thank you very much for your manuscript titled “Nectar robbing by the invasive bumblebee Bombus terrestris (Apidae) changes the behavior of native flower visitors of Fuchsia magellanica Lam. (Onagraceae) but not seed set” that you sent to PeerJ.

This study presents very valuable and relevant information of effects of nectar robbing on nectar production, pollinator foraging behavior, and reproductive success in 24 populations of the plant Fuchsia magellanica.

As you will see below, referee 1's comments suggest major revision, while reviewer 2 suggests minor revision before your paper can be published. Given this, I would like to see a major revision dealing with the comments. Their comments should provide a clear idea for you to review, hopefully improving the clarity and rigor of the presentation of your work. I will be happy to accept your article pending further revisions, detailed by the referees.

Reviewer 1 suggests improving the description of methods, statistical analyses and results.

Reviewer 2 suggests expanding on the introduction and methods, in addition to detailing some points of the discussion.

Please note that we consider these revisions to be important and your revised manuscript will likely need to be revised again.

·

Basic reporting

The manuscript is the right length, written in clear language, and backed with references. Figures and tables are relevant and clear, but could use some editing, especially with descriptions. The raw data provided needs documentation.

Experimental design

The question is defined well, and the motivations for conducting the study are clear. Data collection is rigorous, and the methods are well-described. Some of the methods are unclear, especially the assignment of flowers to treatments, and the side-effects of the treatments on fruit sets. Moreover, I have some questions about the statistical analyses, I describe them later in this report.

Validity of the findings

Conclusions are generally well-supported, with the exception of the treatment of non-recorded values. I detail this in the line-by-line comments, but my comments are more of a question than criticism.

Additional comments

In this paper, Valdivia et al. build upon their experience with flower–visitor communities across temperate forests in Chile. Their previous work documented geographic mosaics in nectar robbing by the introduced bumblebee (*Bombus terrestris*) on the shrub *Fuschia magellanica* (Valdivia et al., 2016). In the present study, they aim to explore the actual effects of these mosaics on the female reproductive success of the plant across 20 populations.

The manuscript presents original research that is the natural progression of long-term work with the system, and is valuable due to the rare documentation of seed-sets in pollination work. It uses an impressive dataset, and I would like to commend the authors on this.

Generally, the manuscript is sound. I have some doubts about the treatment of "non-recorded" nectar data, and therefore the modelling procedures. However, I think that the authors could account for or explain these.

I think that the manuscript is well-organized. I wanted to highlight that I really liked reading the abstract, but I found it more difficult to stay engaged during the Results section, perhaps due to the number-heavy narrative. A suggestion might be to summarise some of the fieldsite-specific data in a table.

The overall story is cohesive, but the background could be filled-in better. Normally, referencing your own work is frowned upon, but in this case it strengthens the authors' case for conducting the presented work. I would encourage them to use their previous work more evidently.

## Abstract
- Line 21: The abstract is written really well! The language is simple and clear, and I enjoyed reading it!
- Line 33-35: I don't think that this direct causation can be concluded from the study design, it is more of a speculation. I would suggest toning this down.
- Line 36-38: This is a broad generalisation, we can only say this of F. magellanica, and then too it would depend on the model clarification from the authors.

## Introduction
- Line 41-42: This isn't technically correct, as not all flowers have hidden reproductive organs. I understand why the authors say this, but I would rephrase to stick to visitation only. Or rephrase it to tubular flowers?
- Line 42: "Effective pollinators" would require knowing pollen transfer rates and seed sets (which I believe the authors do here), but it's too soon to make this point in my opinion. Change to "legitimate"?
- Line 52: Here, it might be worth introducing the reduction in standing crop (as you also test it). Besides this, how about adding the direction of the effect on nectar production rates? Something like "nectar robbing reduces nectar production rates", if that's factually correct.
- Line 54-69: This is a good mini-review, but I am missing a critical reasoning for why we cannot estimate effects of nectar robbing with visitation alone. This might be obvious to someone working with the system, but the general audience might not get why we need to observe seed sets. There needs to be an argument for the idea that pollinator visitation does not always lead to plant reproduction. Then, it will tie-in nicely with the reasoning for the authors statement in line 92-93.
- Line 71-73:
- Is it this specific a combination? There are no other hummingbird and bumblebee species that visit such flowers?
- Missing word, perhaps: ...primarily pollinates these plants which are also...
- Line 73-80: Here, I was most impressed by the long-term work by the authors (and others) on the system. It is a very convincing paragraph!
- Line 86-88: Especially in a manuscript about nectar robbing, it is better to be very careful with terminology, here it should be robbery, as theft does have a specific meaning (as in Irwin 1980 cited above)
- Line 96: typo "rates"

## Methods
- Line 124-135: This is interesting but not directly relevant to the manuscript. Maybe there is something in the discussion that requires it? Perhaps it could help explain why there were some fruits that were not retrieved later? On the other hand, there is no description of the flowers, such as the size and longevity of flowers. Also adding in something about the self-compatibility of the flowers, as this is evident from the results (Line 294-297) would be helpful for readers.
- Line 138-139: Impressive dataset!
- Line 147: What is the size of the capillary tube?
- Line 181-182: This is missing the critical legitimacy definition. I'm sure that the authors considered it, as in the picture, but as it stands, it defines thieving instead of robbing.
- Line 198-199: I'm not sure I follow the design. Were there five flowers per treatment per plant, such that there were 15 flowers per plant? Or were five flowers chosen per plant, and then randomly assigned to one of three categories each, such that each plant had an uneven number of flowers per treatment?
- Line 202-204: The idea is clever, but I wonder whether the masking tape may have unintentionally influenced visitation or fruit set. As the study design is, it would be difficult to say if the -Rob+Pol treatment was affected by the visitors or the masking tape. Unfortunately, there is a second complication with the design. Manipulating the access of certain types of visitors does not mean that the visitation rates are also being manipulated. That is, we do not know if flowers in the -Rob+Pol have the same number of pollinators as the +Rob+Pol treatment. I don't think it can be addressed by the authors unless they have some pilot data to back it up, but I do think that the drawbacks of the design can be clearly discussed in the Discussion.
- Line 216: the abbreviations changed, please standardise them throughout the manuscript.

## Results
- Line 228-242: I feel like this is not directly relevant to the study, in that as readers we know nothing about the populations. It would perhaps be easier if the information in here could be summarised in a table, and directly linked to the linear models where population was among the effects tested.
- Line 240-241: Wouldn't this basically mean that you did measure them, their levels were 0? If yes, the modelling procedure might need to change. This is the main point that I think requires clarification to ensure the validity of the modelling approach.
- Line 283-285: The small effect sizes might have something to do with the scale of the study. Such effects are often studied at the patch scale, while this is at the population-level. I'm not sure that pollinator displacement can be observable, as there would need to be other suitable and accessible areas for the pollinators to move to. Could this be integrated into the Discussion?
## Discussion
- Line 337-344: Although this is logical, it should have some literature to back it up. If there isn't any, are there any field-based observations to support these speculations?
- Line 377-389: I honestly hope that the authors will follow-up on this finding!
- Line 395-400: This statement is bold, but I don't want to evaluate it until the authors clarify what happened with the zero-measures of nectar production rates and standing crop. If their model procedures are sound, this would be valid.

## Figures and Tables
- The figure and table captions need to be improved in places.
- Figure 1
- The summary information in brackets makes the caption tricky to read. Can it be placed into a table, and referenced in the caption? Something like "Each population was numbered in alphabetical order. For details see Table XX"
- Figure 3
- Typo in panel C: terrestris
- Is is true that even for B. terrestris only secondary robbing is plotted?
- Figure 4
- The data and methods behind this figure are unclear. Besides this, as mentioned above, the errors and trends extend below 0 on the Y-axis. Is this correct?
- Figure 5
- isn't referenced in the main text (but I might have missed it)
- Typo on Y-axis: seeds
- Table 2
- change "not" to "no"
- Table 3
- Standardize abbreviations

Reviewer 2 ·

Basic reporting

The manuscript "Nectar robbing by the invasive bumblebee Bombus terrestris (Apidae) changes the behavior of native flower visitors of Fuchsia magellanica Lam. (Onagraceae) but not seed set" is a detailed ecological study of the relationships between an emblematic plant species, its pollinators and nectar robbers; it also evaluates the effect of an exotic insect species (that has been introduced widely around the world) on the viability of the population.

The manuscript is well-structured, and the language is correct and easy to understand. The questions posed are addressed with data. There are some very minor comments on the annotated manuscript and in the "additional comments" section. The literature references are ok. The raw data are shared and the figures are appropriate. The legend of Fig. 5 repeats that of Fig. 4, probably because of an error.

While the introduction includes several relevant subject, I suggest explaining why the Fuchsia magellanica was selected as a model and add some information on its general distribution, ecology (particularly its reproductive system) and uses; the same goes for the introduced Bombus. The intro should highlight more that the work addresses the effect of an introduced species on a native ecosystem and very briefly review how this and other introduced insects that may alter the pollination ecology of native species. The native forest should also be described briefly.

Experimental design

The manuscript is well within the scope of the journal. The research questions are clearly stated and relevant. Various methods and approaches are used. It is based in a geographically disperse, large dataset.

The methods are generally well-described, but I I suggest being more explicit on the criteria of the site, individual plant, branch and flower selection (if it was not with some sort of a lottery, but preferential, that should be stated). And how did you know there were varying levels of primary nectar robbing by B. terrestris before starting the study? You probably had observations or previous data, but this should be stated. Perhaps you could indicate if all sites had B. terrestris (the results show that there was no nectar robbing at some sites, but you do not state if this was due to B. terrestris being absent); perhaps use two colors on the map?

Please define what you mean by "standing crop", as the meaning (average amount of nectar in a flower) is no immediately understandable; a reader unfamiliar with the literature might interpret it as all of the nectar of a plant or population.

Please state how taxonomic identity of the plants and animals were ensured (voucher specimens? Specialists?). Also, did the observer(s) have previous training with the possible species? What was the distance between observer and observed flower? Did the hummingbirds react to the presence of the observer?

The authors mention permits from their institutions; as this is an observational study, further permits may have been unnecessary.

Validity of the findings

The data are provided and appear to be solid, if with much variation. The conclusions are appropriate. Though the results are negative to an extent, I think the work contributes substantially to the knowledge and theory of pollination ecology, and should be published after some improvements.

The variation in seed set between different populations, even under similar treatments, was rather large. Any interpretation that goes further than generic "environmental factors"?

In the Discussion, I suggest expanding the interpretation of both the influence of the reproductive system and of the influence of other factors (rain, sun, fertility, etc.) on nectar production. And the results could also be interpreted as pollinators not being a limiting factor for seed set in this species, or not?

Additional comments

There are some minor stylistic issues where the language could be more concise (e.g. in the abstract: "However, in some populations, it is also affected by nectar robbing ..." could be "Some populations are also affected by nectar robbing ..."; there are several such instances and the authors might want have their manuscript reviewed again by someone good with scientific editing in English. However, these few instances do not distract substantially for the text, so I leave it up to the authors.

The authors may want to think about providing a shorter, catchy title, but I leave this up to them. Also, some paragraphs are somewhat long and could be separated to improve readability.

Please note that the annotated manuscript was made with Foxit, but it should be readable in Adobe.

Annotated reviews are not available for download in order to protect the identity of reviewers who chose to remain anonymous.

---

## Round 0.2 · accepted · Accept

After reviewing this revised version of your manuscript, I see that the main comments suggested by the reviewers have been included, while the suggestions not considered are justified in detail. Therefore, I am satisfied with the current version and consider it ready for publication.

·

Basic reporting

The revised article is clear, easy to follow. I am satisfied with the authors' answers to my questions and how they implemented my suggestions. I found the comments raised by reviewer #2 to also be useful and addressed by the authors.

Experimental design

No comment, the authors have explained their methods better in the new version, and elaborated on the modelling approach.

Validity of the findings

No comment, the authors have explained the modelling procedure which clears my previous comments.